# Asthma status and suicidal behavior risk: A meta-analysis of cohort studies

**Wei He** , Zhixin You, Huijiao Li, Yiping Wang, Guohua Li*

Department of Respiratory, Shaoxing Hospital of Traditional Chinese Medicine Affiliated to Zhejiang Chinese Medical University, Shaoxing, Zhejiang, China

* 745815167@qq.com

## Abstract

### Objective

This meta-analysis investigates the differential risks of suicidal behaviors (ideation, attempts, mortality) associated with current asthma and asthma history.

### Method

Retrieve cohort studies on the association between asthma and suicide from PubMed, Embase, and Cochrane library database. Use the Newcastle Ottawa Quality Assessment Scale (NOS) to assess the risk of bias. The risk ratio (RR) of 95% confidence interval (CI) was summarized using a random effects model, and publication bias was evaluated using funnel plots and Egger's trials.

### Result

A total of 12 cohort studies were included and published between 2005 and 2024. The NOS scores for the 12 cohort studies included in this meta-analysis ranged from 7 to 9. Most studies received scores of 7 or 8, indicating a generally high quality. Current asthma conferred a 62% increased risk of suicidal behaviors (RR = 1.62, 95% CI: 1.38–1.88), with suicide attempts showing the strongest association (RR = 2.27, 95% CI: 1.33–3.89). Asthma history was linked only to elevated suicide mortality (RR = 1.87, 95% CI: 1.64–2.14), not non-fatal suicidal behaviors.

### Conclusion

Current asthma status is associated with an increased risk of suicidal behaviors, but a history of asthma correlates only with elevated suicide mortality. This finding highlights the need for proactive mental health screening in asthma management protocols, especially during periods of active disease.

**Data availability statement:** All relevant data are within the manuscript and its Supporting Information files.

**Funding:** The author(s) received no specific funding for this work.

**Competing interests:** The authors have declared that no competing interests exist.

## Introduction

Asthma is a chronic respiratory disease characterized by airway inflammation, causing symptoms like coughing and wheezing [1]. It affects over 300 million people globally, with significant morbidity and mortality [2]. Despite decreasing age-standardized rates, the overall burden remains severe, with 262.41 million prevalent cases in 2019 [3]. Asthma is heterogeneous, with various phenotypes and endotypes identified in recent years [4]. Risk factors include high body mass index, occupational exposures, and smoking [5]. The disease burden is particularly concerning in low-income countries and among children [6]. Asthma may be a risk factor for certain respiratory diseases, including chronic obstructive pulmonary disease (COPD) [7], lung cancer [8]. In addition, recent meta-analyses and cohort studies suggest an association between asthma and increased risk of psychiatric disorders. A systematic review found higher prevalence of anxiety disorders in youth with chronic medical conditions, including asthma, compared to the general population [9]. Mental disorders, including asthma-related conditions like atopic dermatitis, were associated with higher suicide risk [10] and increased odds of internalizing behaviors and depression symptoms in children [11].

Suicide is a major public health problem with significant global burden [12]. Recent research has explored various risk factors for suicidal behavior, including exposure to suicide or suicide attempts [13], air pollution [14,15], and infectious disease epidemics [16]. Studies have also investigated the association between suicide and inflammation markers like C-reactive protein [17]. And its specific association with suicidal behavior remains unclear. Therefore, we conducted a meta-analysis to further explore the association between different states of asthma and suicidal behavior.

## Methods

This meta-analysis was conducted in accordance with the updated Preferred Reporting Items for Systematic Reviews and Meta-Analyses (PRISMA 2020) guidelines [18]. The protocol for the study was registered on the PROSPERO platform (CRD42025635002).

### Data sources

A comprehensive literature search was performed across PubMed, Embase, and the Cochrane Library to identify relevant studies published up until January 5, 2025. The search strategy combined both controlled vocabulary and keywords, using terms such as "suicide," "asthma," and their respective synonyms. We employed MeSH (Medical Subject Headings) terms in PubMed and Emtree terms in Embase to ensure a thorough search. Additionally, the reference lists of retrieved studies and previous meta-analyses were reviewed to identify potentially eligible studies. A detailed search strategy for each database is provided in Supplementary Tables 1–3 in S1 File.

### Eligibility criteria

Studies were considered eligible if they met the following criteria: (a) Participants: Individuals diagnosed with asthma who exhibited suicidal behavior, including suicide ideation, suicide attempts, or suicide mortality; (b) Exposure: Patients with asthma,

focusing on those with either current asthma, an asthma attack, or a history of asthma; (c) Comparator: Healthy individuals or those without asthma; (d) Outcome: Reported incidents of suicidal behaviors, with available effect sizes such as hazard ratios (HR), relative risks (RR), or odds ratios (OR), including 95% confidence intervals (CIs). Only studies reporting effect sizes adjusted for potential confounders were included; (e) Study Design: Cohort studies (prospective or retrospective) or cohort-based nested case-control studies; (f) no language or regional restrictions were applied in the selection of studies to ensure a comprehensive and global representation of the data.

Exclusion criteria included: (a) Conference abstracts or letters to the editor; (b) Studies with incomplete data or lacking the relevant outcomes of interest; (c) Population or exposure combined with other diseases.

### Study selection

The initial records identified in the search were imported into NoteExpress reference management software, where duplicates were removed. Two authors (W He, ZX You) independently screened the titles and abstracts to exclude irrelevant studies. The remaining studies were further assessed for eligibility based on predefined inclusion and exclusion criteria. For studies with uncertain eligibility, the full text was reviewed. Any discrepancies were resolved through group discussion.

### Data extraction

Data extraction was carried out using a custom-designed form in Excel (Microsoft Corporation, USA). Two authors (HJ Li, YP Wang) independently extracted data from eligible studies. Key information, including first author, publication year, country, data source, and adjusted confounders, was collected. The extracted data were cross-checked, and any discrepancies were resolved through discussion.

### Study quality

The quality of the included studies was assessed using the Newcastle-Ottawa Quality Assessment Scale (NOS) (available at http://www.ohri.ca/programs/clinical_epidemiology/oxford.asp). The assessment focused on three domains: selection, comparability, and outcomes. Cohort studies were assigned scores ranging from 0 to 9 stars, with higher scores indicating better study quality. Studies were categorized as high (7–9 stars), moderate (4–6 stars), or low quality (0–3 stars).

### Data synthesis

Data analysis was conducted using Stata software (version 14). Given the potential clinical and methodological heterogeneity, we selected a random effects model for the meta-analysis to ensure the robustness and generalizability of the results [19,20]. Sensitivity analysis was performed to test the stability of the overall findings and to investigate sources of heterogeneity. We also conducted independent subgroup analyses based on asthma status and different types of suicidal behaviors. Publication bias was assessed using funnel plots and Egger's tests.

## Results

### Study selection

A total of 721 original records were identified. After removing duplicates and screening titles and abstracts, 22 articles were selected for full-text review. Following the full-text assessment, 12 cohort studies were included in this meta-analysis [21–32]. The exclusion criteria involved 1 conference abstract [33], 6 non-cohort studies [34–39], and 3 studies that investigated multiple diseases [40–42]. The study selection process is shown in Fig 1.

### Study characteristics and quality

A total of 12 cohort studies, spanning prospective and retrospective designs, were included in this meta-analysis. These studies, conducted in countries such as Korea, China, the USA, Canada, Denmark, and the UK, utilized diverse data

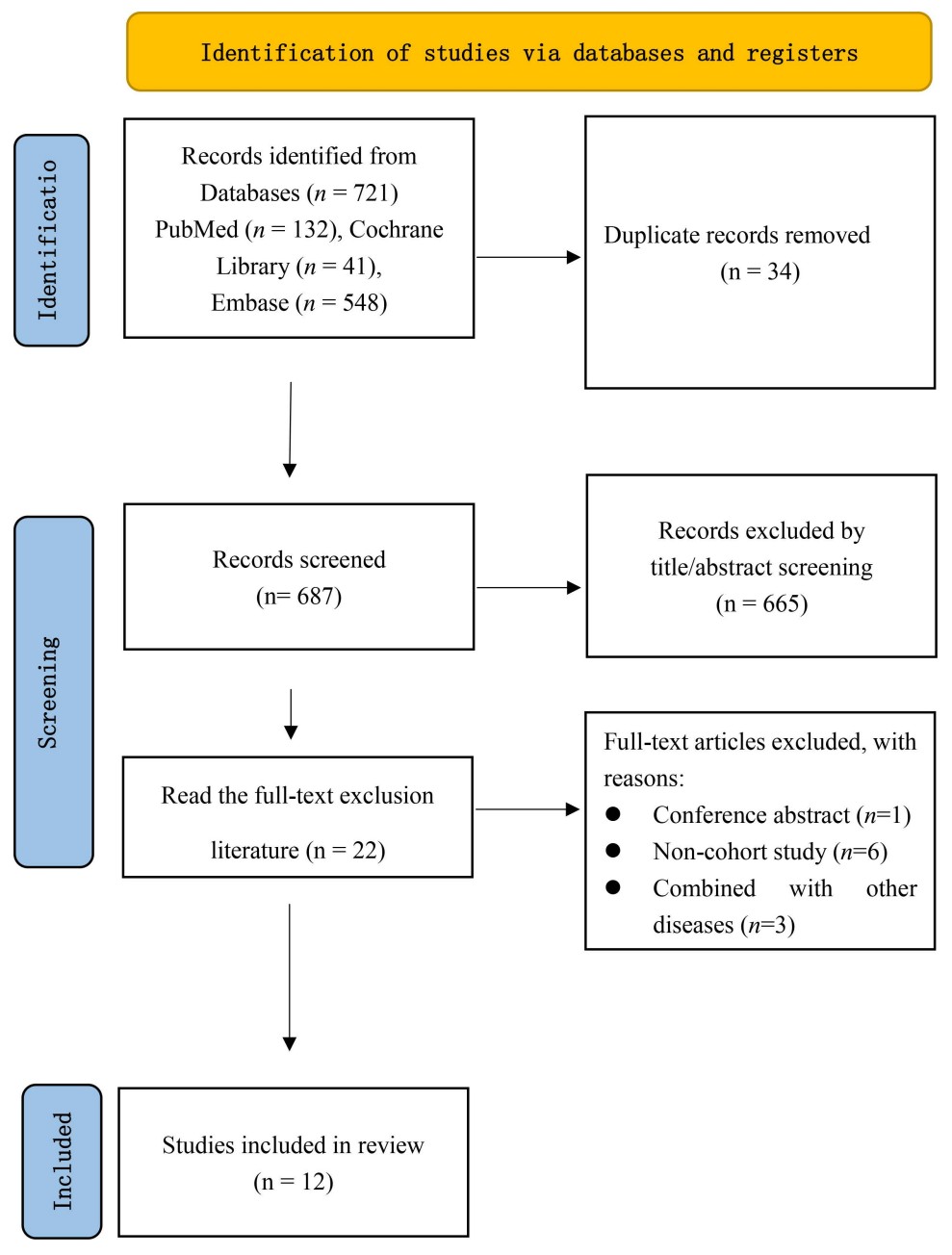

**Fig 1. Literature screening process.**

sources, including national health insurance databases and health surveys. Sample sizes ranged from 830 to 7,140,589, with a total of over 12 million participants. The baseline age of participants varied from 5 to 65 years, with follow-up periods ranging from 2 to 20 years. Suicidal behavior events ranged from 32 to 8,721 across studies, involving both general populations and individuals with conditions like asthma and mental health disorders. Confounder adjustments included age, sex, socioeconomic status, smoking, alcohol use, and mental health conditions. The NOS scores for the 12 cohort studies included in this meta-analysis ranged from 7 to 9. Most studies received scores of 7 or 8, indicating a generally high quality. The characteristics and quality of included cohort studies are summarized in Table 1.

**Table 1. Basic characteristics of included cohort studies.**

| Author | Year | Country | Study type | Data source | Baseline Age | Follow-up time | Sample size | Events | Confounder Adjustment | NOS score |
|---|---|---|---|---|---|---|---|---|---|---|
| Kim | 2024 | Korea | Prospective cohort | Korean National Health Insurance Service data | ≥ 20 | 12.3 years | 3,914,041 | 1,383 | / | 7 |
| Chen | 2024 | China | Retrospective cohort | National Health Insurance Research Database | 11-16 | 20 years | 153,526 | 285 | Sex, age, cigarette smoking by at least one family member, allergy and cigarette smoking, anxiety, bipolar, eating disorder, personality disorder, schizophrenia, alcohol, depression | 8 |
| Hoffman | 2022 | USA | Prospective cohort | Adolescent Brain Cognitive Development (ABCD) Study | 9 - 12 | 2 years | 11,876 | 522 | Age, sex, race (Black, White, Other) and Hispanic ethnicity, demographics, household-level socioeconomic factors (income, average parent education, and maternal age), and neighborhood-level factors (area deprivation index, population density, NO, PM 2.5, and proximity to major roads),BPM score | 7 |
| Huh | 2021 | Korea | Prospective cohort | Korea National Health and Nutrition Examination Survey (KNHANES), | 46.3(Mean) | 4 years | 830 | 63 | Age, sex, educational level, personal income, smoking status, alcohol consumption, body mass index, comorbidity, and depressive mood | 7 |
| Bolton | 2015 | Canada | Retrospective cohort | Population Health Research Data Repository at the Manitoba Centre for Health Policy, within the University of Manitoba | / | 13 years | 8400 | 418 | Aggregated Diagnosis Group (ADG) count, depression, anxiety disorders, substance abuse or dependence, schizophrenia, dementia, other psychosocial disorders, and all physical disorders simultaneously | 7 |
| Crawford | 2015 | UK | Prospective cohort | National Health Service Central Register (NHSCR) c | 19.4(Mean) | 11.8 years | 11,463 | 32 | Sex, height, number of siblings, birth order, fathers social class, body mass index and current smoking | 7 |
| Crump | 2014 | USA | Prospective cohort | Swedish Death Registry maintained by the National Board of Health and Welfare | ≥ 18 | 8 years | 7,140,589 | 8721 | Age, marital status, country of birth, education, employment status, income and urban/rural status, psychiatric and somatic disorders | 8 |
| Singhal | 2014 | UK | Retrospective cohort | Hospital Episode Statistics (HES)of National Health Service (NHS) | 10 - 65 | 12 years | 31,700 | 662 | Age in five-year bands, time period in single calendar years, region of residence and deprivation score associated with patients' area of residence, in quintiles | 9 |
| Schumock | 2012 | USA | Retrospective cohort | LifeLink Health Plan Claims Database | 5 - 24 | 10 years | 3,438 | 344 | Current users of LTMAs compared with never users. | 7 |
| Christiansen | 2012 | Denmark | Retrospective cohort | Danish Civil Register | ≤ 18 | 16 years | 14,760 | 738 | Child's psychiatric history, parents' psychiatric history, use of psychopharmacological drugs, level of income, level of education, death and cohabitation. | 7 |
| Kuo | 2010 | China | Prospective cohort | Department of Health Death Certification System in Taiwan | 11 - 16 | 12 years | 30,132 | 844 | Gender, age, cigarette smoking by at least one member of the family, cigarette smoking, and allergic rhinitis. | 8 |
| Goodwin | 2005 | USA | Retrospective cohort | Baltimore Epidemiologic Catchment Area Study | ≥ 18 | 13 years | 3504 | 318 | Gender, age, race, treatment for asthma, and major depression (lifetime) | 8 |

## Meta-analysis of current asthma and suicidal behavior risk

A total of 7 cohorts reported the association between current asthma status and various suicidal behaviors. The meta-analysis revealed an overall increased risk of suicidal behaviors associated with current asthma (RR = 1.62, 95% CI: 1.38–1.88, $I^2$ = 74.3%, Fig 2). For specific suicidal behaviors, the meta-analysis of 4 cohorts showed that current asthma was linked to an increased risk of suicide mortality (RR = 1.58, 95% CI: 1.26–1.98, $I^2$ = 82.1%, Fig 2); 3 cohorts indicated an increased risk of suicidal ideation (RR = 1.69, 95% CI: 1.10–2.59, $I^2$ = 72.8%, Fig 2); and 2 cohorts suggested an elevated risk of suicide attempts (RR = 2.27, 95% CI: 1.33–3.89, $I^2$ = 24.3%, Fig 2).

Sensitivity analysis using the one-by-one elimination method showed that no single study significantly altered the overall results, indicating good stability of the meta-analysis findings (Supplementary Fig 1 in S1 File).

The funnel plot visually indicated symmetry between current asthma status and the risk of suicidal behavior. However, the Egger test (p = 0.001) suggested the presence of publication bias, which was corrected using the "Trim and Fill Analysis". The adjusted funnel plot is shown in Supplementary Fig 2 in S1 File.

## Meta-analysis of asthma history and suicidal behavior risk

A total of 8 cohorts examined the association between a history of asthma and various suicidal behaviors. Meta-analyses demonstrated an overall increased risk of suicidal behaviors in patients with a history of asthma (RR = 1.43, 95% CI: 1.12–1.83, $I^2$ = 91.0%, Fig 3). For specific suicidal behaviors, including suicide ideation, suicide attempts, and suicide mortality, the meta-analysis of 4 cohorts showed that an increased risk of suicide mortality among patients with a history of

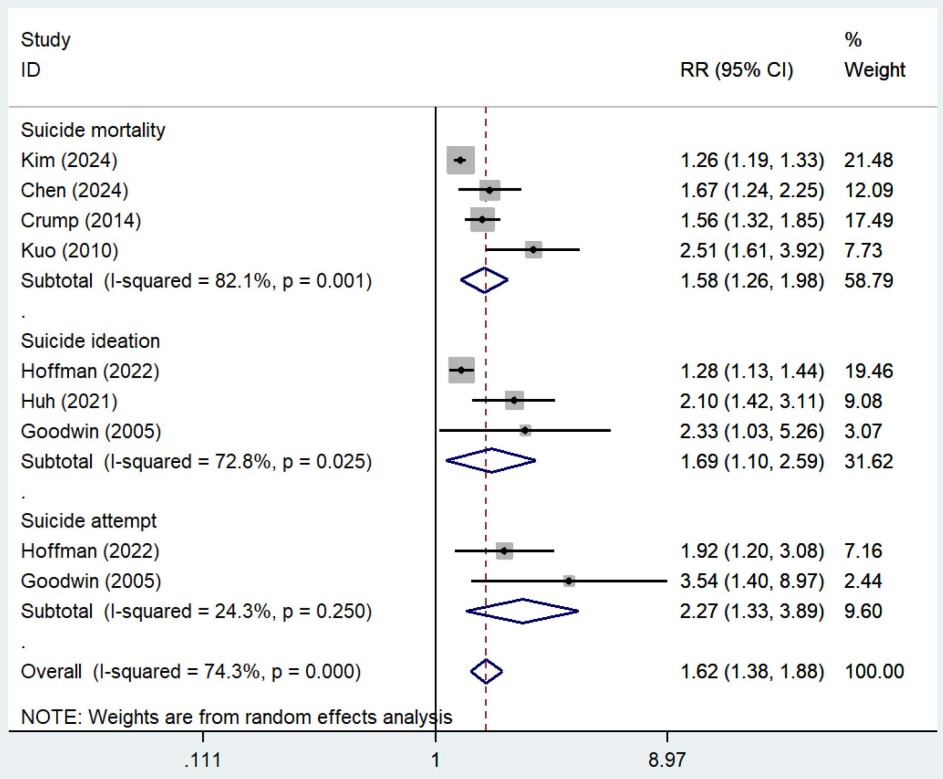

**Fig 2. Current asthma and suicidal behavior risk.**

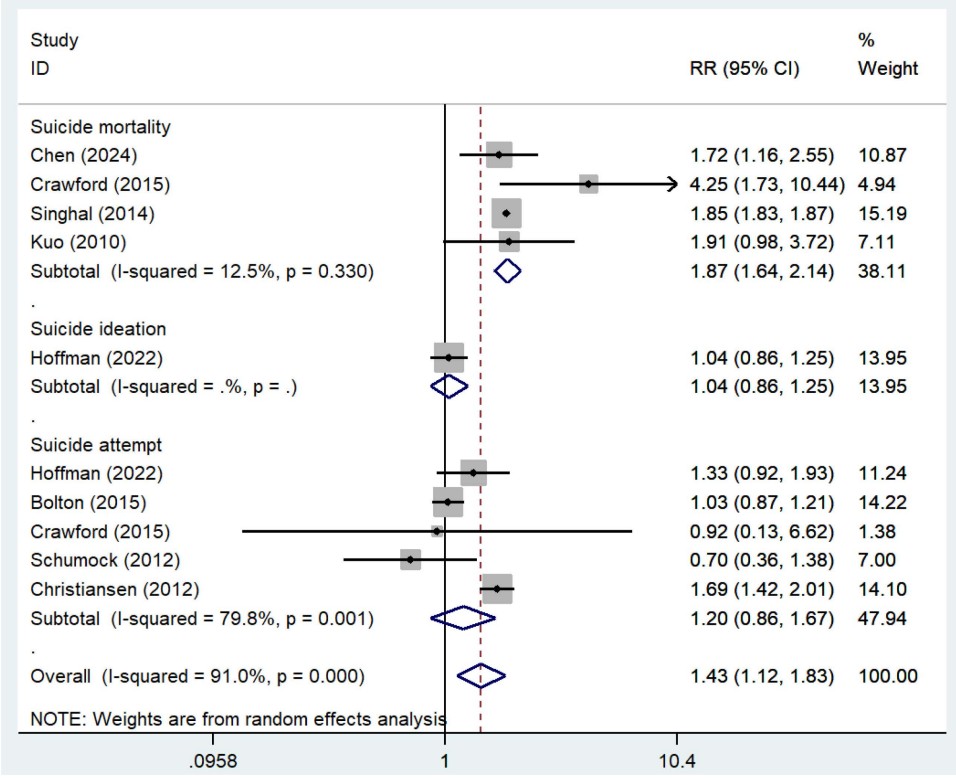

**Fig 3. Asthma history and suicidal behavior risk.**

asthma (RR = 1.87, 95% CI: 1.64–2.14, I² = 12.5%, Fig 3). One cohort showed no significant association between a history of asthma and suicidal ideation (RR = 1.04, 95% CI: 0.86–1.25, Fig 3). Additionally, meta-analyses of 5 cohorts indicated no significant association between a history of asthma and suicidal attempts (RR = 1.20, 95% CI: 0.86–1.67, I² = 79.8%, Fig 3).

Sensitivity analysis using the single-study elimination method demonstrated that no individual study significantly altered the overall results, suggesting good stability of the meta-analysis findings (Supplementary Fig 3 in S1 File).

The funnel plot visually indicated symmetry between a history of asthma and the risk of suicidal behavior. Furthermore, the Egger test (p = 0.1) did not provide evidence of publication bias, as shown in the funnel plot in Supplementary Fig 4 in S1 File.

## Discussion

### Principal findings

This meta-analysis of 12 cohort studies involving over 12 million participants demonstrates that individuals with current asthma face a significantly elevated risk of suicidal behaviors (RR = 1.62, 95% CI: 1.38–1.88), particularly suicide attempts (RR = 2.27) and mortality (RR = 1.58). In contrast, a history of asthma was associated with increased suicide mortality (RR = 1.87) but not with suicidal ideation or attempts. The heterogeneity observed (I² = 74.3–91.0%) likely stems from variations in study designs, population demographics, and adjustments for confounders such as mental health comorbidities and socioeconomic factors. These findings underscore the importance of distinguishing between current and historical asthma status when evaluating suicide risk.

## Comparison with previous studies

Our results align with prior studies linking chronic inflammatory conditions to psychiatric morbidity and suicidal behavior. A 2021 meta-analysis by Moitra et al. [43] reported that atopic diseases, including asthma, were associated with a 30% increased suicide risk (OR=1.30, 95% CI: 1.16–1.47). However, our study extends this evidence by stratifying risks by asthma status and specific suicidal outcomes. Unlike their pooled analysis of suicidality outcomes, our study uniquely stratifies risks by asthma status, revealing a 62% increased risk for current asthma—particularly for suicide attempts, a dimension not previously disaggregated—and identifying elevated suicide mortality even among those with resolved asthma. Furthermore, we address temporal and geographic limitations in prior work by incorporating seven recent cohorts (2020–2024) and diverse populations, including Asian cohorts underrepresented in earlier analyses. Crucially, our rigorous adjustment for mental health comorbidities and socioeconomic confounders, inconsistently addressed in earlier studies, strengthens causal inference and explains higher risk estimates compared to Zhang et al.'s pooled odds ratio [44]. These advancements underscore the necessity of dynamic, status-specific risk assessment in clinical practice rather than relying on lifetime asthma diagnosis alone.

## Interpretation of findings

The increased suicide risk associated with current asthma may stem from both biological and psychosocial mechanisms. Chronic airway inflammation in active asthma may exacerbate neuroinflammation, disrupt hypothalamic-pituitary-adrenal (HPA) axis activity, and elevate pro-inflammatory cytokines such as IL-6, which are known to be involved in depression and impulsivity [45,46]. Furthermore, asthma-related sleep disturbances, the side effects of corticosteroids (e.g., mood swings), and the psychological strain of recurrent exacerbations may all contribute to suicidal behavior. The absence of a significant association between asthma history and non-fatal suicidal outcomes suggests that resolved inflammation or improved disease management might reduce these risks. Several studies have shown that asthma is associated with a higher likelihood of suicidal ideation and attempts, even after adjusting for comorbid mental health disorders. This association appears to be stronger in individuals with more severe and persistent asthma symptoms [31]. Moreover, social factors such as bullying may amplify the relationship between asthma and suicidal behaviors, particularly in adolescents [47]. These findings highlight the critical need for mental health screening and care for asthma patients, especially for younger individuals with severe symptoms [48].

## Strengths and limitations

Clinically, these findings advocate for integrated care models that screen asthma patients—particularly those with active disease—for suicidal ideation and mental health comorbidities. Public health strategies should prioritize reducing asthma-related stigma and improving access to psychosocial support in high-risk populations. This study has limitations. First, due to significant variation in baseline age across the included studies, we were unable to perform age-adjusted analyses. Age is an important factor in suicidal behavior, and future studies should adjust for age to better understand its impact on the relationship between asthma and suicide risk. Second, we observed some evidence of publication bias, the quality of the included studies was generally high, and sensitivity analyses were performed to assess the robustness of the results. However, we acknowledge that publication bias may influence the overall conclusions, particularly given the small sample size of the included studies (n = 12). Lastly, it is important to note that while the included studies did not consistently report on asthma severity, anecdotal evidence and previous research suggest that severe asthma may be more strongly associated with negative mental health outcomes. Further studies should explore this dimension.

## Conclusions

Current asthma status is associated with a 62% increased risk of suicidal behaviors, with suicide attempts showing the strongest association. In contrast, a history of asthma correlates only with elevated suicide mortality. These results

highlight the need for proactive mental health screening in asthma management protocols, especially during periods of active disease.

## Supporting information

**S1 File.   Tables 1–3: The literature search steps and results of Pubmed, Embase and Cochrane library. Figure 1: Sensitivity analysis of current asthma and suicidal behavior risk. Figure 2: Funnel plot of current asthma and suicidal behavior risk. Figure 3: Sensitivity analysis of asthma history and suicidal behavior risk. Figure 4: Funnel plot of asthma history and suicidal behavior risk.**
(DOCX)

**S2 File.   Data transparency.** Sheet 1: All 721 records obtained from the database. Sheet 2: List of excluded documents. Sheet 3: Data extraction quality assess. Sheet 4: Data analysis process.
(XLS)

**S1 Checklist.   PRISMA checklist.**
(DOCX)

## Author contributions

**Conceptualization:** Wei He, Guohua Li.

**Data curation:** Wei He, Zhixin You, Huijiao Li, Yiping Wang.

**Formal analysis:** Zhixin You.

**Funding acquisition:** Guohua Li.

**Methodology:** Huijiao Li, Yiping Wang.

**Software:** Zhixin You, Huijiao Li, Yiping Wang.

**Writing – original draft:** Wei He, Yiping Wang, Guohua Li.

**Writing – review & editing:** Wei He, Guohua Li.

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
