## [Decision Letter · Decision Letter 0]

25 Mar 2025

PONE-D-25-08475Asthma Status and Suicidal Behavior Risk: A Meta-Analysis of Cohort StudiesPLOS ONE

Dear Dr. He,

Thank you for submitting your manuscript to PLOS ONE. After careful consideration, we feel that it has merit but does not fully meet PLOS ONE’s publication criteria as it currently stands. Therefore, we invite you to submit a revised version of the manuscript that addresses the points raised during the review process.

The metanalysis fits the criteria for publication in the journal with mostly sound methodology and well presented results. The manuscript will benefit from refinement based on the reviewer's comments including clarifying in the inclusion criteria the definition of Asthma, some subsections clarifications in the methodology with inclusion of the PRISMA checklist and expanding the discussion further on interpretation of the results and implications for practice.

We look forward to receiving your revised manuscript.

Kind regards,

Toufic Ahmad Chaaban, M.D.

Academic Editor

PLOS ONE

Journal Requirements:

2. As required by our policy on Data Availability, please ensure your manuscript or supplementary information includes the following:

5. Please remove all personal information, ensure that the data shared are in accordance with participant consent, and re-upload a fully anonymized data set.

Additional Editor Comments :

The metanalysis fits the criteria for publication in the journal with mostly sound methodology and well presented results. The manuscript will benefit from refinement based on the reviewer's comments including clarifying in the inclusion criteria the definition of Asthma, some subsections clarifications in the methodology with inclusion of the PRISMA checklist and expanding the discussion further on interpretation of the results and implications for practice.

Reviewers' comments:

Reviewer's Responses to Questions

**Comments to the Author**

1. Is the manuscript technically sound, and do the data support the conclusions?

Reviewer #1: Yes

Reviewer #2: Partly

Reviewer #3: Yes

2. Has the statistical analysis been performed appropriately and rigorously? 

Reviewer #1: Yes

Reviewer #2: Yes

Reviewer #3: Yes

3. Have the authors made all data underlying the findings in their manuscript fully available?

Reviewer #1: Yes

Reviewer #2: No

Reviewer #3: Yes

4. Is the manuscript presented in an intelligible fashion and written in standard English?

Reviewer #1: Yes

Reviewer #2: Yes

Reviewer #3: Yes

5. Review Comments to the Author

Reviewer #1: Thank you for the opportunity to review your manuscript, which is on an interesting and important aspect of asthma care. The manuscript is generally well written and is engaging.

Reading through the manuscript, I had the following queries, which I feel should be addressed:

1) Page 3, Lines 6-8, I would advise removing the reference to asthma's relation to COVID-19 mortality, as it is not relevant to this study

2) It would be useful to have some information on comment on how the authors of the included studies have defined asthma, and particularly whether there was any discussion of severe asthma vs controlled asthma. Anecdotally, we have found severe asthma is particularly associated with negative effects on mental health, and it would be good for authors to comment on whether the papers included looked at asthma severity

3) There is significant heterogeneity in terms of baseline age of patients - did the authors do any work looking at age-adjusted mortality/suicide risk? Often suicidal behaviour will vary in age groups, with (at least in the UK) an increase in middle age, which some studies did not capture

4) Page 12, line 4 - Authors state "For specific suicidal behaviours, the meta-analysis of 4 cohorts showed that...". Could the authors be clearer on what the specific suicidal behaviours are or what they mean by this? It is unclear to my reading

5) Page 14, line 18 - Authors state they propose IL-6 dysregulation as a potential contributor for these findings, but this is actually outlined later in the paper, and could be restructured. The authors state there is post-2019 evidence but do not cite at this point.

Reviewer #2: Overall, this paper is important. I appreciate the work the authors put into compiling the studies in this systematic review.

General Comments:

— Given the similar study by Zhang et al. (2019), titled 'Suicidality among patients with asthma: A systematic review and meta-analysis,' could the authors clarify the added value of this study to the existing literature? Are there any new insights, differences in methodology, or unique findings that distinguish this study from the previous studies?

Specific Line-by-Line Comments

— Page 4, subsection Eligibility criteria, first line of the second paragraph: The statement "Exclusion criteria included: (b) Duplicate publications" may not be accurate. Removing duplicates is a data management step, not an exclusion criterion. It would be clearer to describe this in the study selection process.

— Same subsection (Eligibility criteria): Were language and region considered as criteria? Please clarify whether language or regional restrictions were applied. These factors affect how complete and useful the review is.

— Page 5, Subsection Study Quality: The authors stated that they used three domains from the Newcastle-Ottawa Scale (NOS): selection, comparability, and outcomes. These domains appear to cover the main aspects of study quality. However, given the focus on social behavior and asthma, do you think these three domains are sufficient to assess the quality of the included studies? Additional factors related to social and environmental influences may also be relevant. I would appreciate knowing whether any other factors should be considered alongside these three domains to ensure a comprehensive quality assessment.

— Page 12, Subsection Meta-analysis of current asthma and suicidal behavior risk: The authors suggested the presence of publication bias. Given that publication bias may lead to an incomplete or skewed representation of the evidence, can the quality of the included studies still be considered high? It is important to address how this potential bias might influence the overall quality and conclusions of the review, particularly considering the small sample size of included studies (only 12).

— The PRISMA 2020 for Abstracts checklist is not attached. Why?

— The authors have stated that all data are fully available without restriction. However, I was unable to locate any attached data or a link to access it. Could the authors please clarify where the data can be accessed or provide the appropriate link to ensure transparency and reproducibility?

Reviewer #3: This is a great and interesting meta-analysis exploring the relationship between asthma and suicidal behaviours. The paper was easy to follow and generally well written. I only have 1 main recommendation:

1) I would recommend going into more depth regarding the relationship between asthma and suicidal behaviours as it was a bit unclear in this paper. Specifically, the section in the discussion, 'Interpretation of findings', should be expanded. Rather than listing general reasons for the relationship between asthma and suicidal behaviours, the authors should delve deeper into the specific mechanisms, and how they play a role in suicidal behaviors, and whether their findings support any of these mechanisms. Additionally, the last line of this section mentions "However, residual confounding by unmeasured factors (e.g., childhood trauma, genetic susceptibility) cannot be excluded(48)." - I wonder if the findings from this meta analysis is able to provide more information on what other confounding factors may bias this finding and how.

Perhaps also highlighting why asthma and suicidal behaviours is related in the Introduction would help the reader see that this association could potentially be more than a correlation.

6. PLOS authors have the option to publish the peer review history of their article (what does this mean? ). If published, this will include your full peer review and any attached files.

**Do you want your identity to be public for this peer review?** For information about this choice, including consent withdrawal, please see our Privacy Policy .

Reviewer #1: No

Reviewer #2: No

Reviewer #3: No

---

## [Author Response · Author response to Decision Letter 1]

3 Apr 2025

Response to reviewers and editors

Dear reviewers and editors:

On behalf of my co-authors, we thank you very much for your letter and comments on our manuscript entitled " Asthma Status and Suicidal Behavior Risk: A Meta-Analysis of Cohort Studies" (Manuscript ID: PONE-D-25-08475).

We appreciate the editors and reviewers for their constructive and valuable comments.

We have revised our manuscript considerably according to the editors’ and reviewers’ comments, questions, and suggestions. In the event that we missed any one of the comments please let us know. This document includes our responses to reviewers and editor comments point by point, and the revised portion are marked in Red in our manuscript.

Reply to reviewers 2

Reply to editorial comments 6

Reply to reviewer 1

Comment 1: Thank you for the opportunity to review your manuscript, which is on an interesting and important aspect of asthma care. The manuscript is generally well written and is engaging.

Reading through the manuscript, I had the following queries, which I feel should be addressed.

Reply 1: Thank you for your positive and encouraging comments on our manuscript. We appreciate your thoughtful feedback and will address your concerns in the following responses.

Comment 2: Page 3, Lines 6-8, I would advise removing the reference to asthma's relation to COVID-19 mortality, as it is not relevant to this study.

Reply 2: Thank you for your suggestion. We agree that the reference to asthma’s relationship with COVID-19 mortality is not central to the scope of this study. We have removed this reference from the manuscript to ensure that the focus remains on the primary subject matter.

Changes in the text: The reference to asthma and COVID-19 mortality on page 3, lines 6-8, has been removed.

Comment 3: It would be useful to have some information on comment on how the authors of the included studies have defined asthma, and particularly whether there was any discussion of severe asthma vs controlled asthma. Anecdotally, we have found severe asthma is particularly associated with negative effects on mental health, and it would be good for authors to comment on whether the papers included looked at asthma severity.

Reply 3: Thank you for your insightful suggestion. We acknowledge that asthma severity is an important factor that may influence mental health outcomes. While the studies included in our meta-analysis did not uniformly report on asthma severity, we have added a brief comment in the manuscript regarding the general lack of detailed classification of asthma severity in the studies. We also note the potential relevance of severe asthma and its possible association with negative mental health outcomes. Future studies should consider incorporating asthma severity as a key variable.

Changes in the text: We have added the following text in the Strengths and limitations section: " Lastly, it is important to note that while the included studies did not consistently report on asthma severity, anecdotal evidence and previous research suggest that severe asthma may be more strongly associated with negative mental health outcomes. Further studies should explore this dimension. "(Page 15, Line 3~6 in red)

Comment 4: There is significant heterogeneity in terms of baseline age of patients - did the authors do any work looking at age-adjusted mortality/suicide risk? Often suicidal behavior will vary in age groups, with (at least in the UK) an increase in middle age, which some studies did not capture

Reply 4: Thank you for your valuable and very professional advice. In conducting this meta-analysis, we also considered that age differences and age overlaps between studies might influence suicidal behavior. Fortunately, most of the included cohorts controlled for age as a possible confounder, so our results are robust. In response to your comments, we briefly mentioned the potential effect of age in the Strengths and limitations section, noting that changes in baseline age in the study may have contributed to heterogeneity. In addition, we recommend that future studies consider adjusting for age when examining the relationship between asthma and suicidal behavior.

Changes in the text: We have added the following text in the limitations section: " First, due to significant variation in baseline age across the included studies, we were unable to perform age-adjusted analyses. Age is an important factor in suicidal behavior, and future studies should adjust for age to better understand its impact on the relationship between asthma and suicide risk. " (Page 15~16, Line 21-22, 1-2 in red)

Comment 5: Page 12, line 4 - Authors state "For specific suicidal behaviours, the meta-analysis of 4 cohorts showed that...". Could the authors be clearer on what the specific suicidal behaviours are or what they mean by this? It is unclear to my reading

Reply 5: Thank you for pointing this out. We have clarified the types of suicidal behaviors referenced in this sentence. The term "specific suicidal behaviors" refers to suicide ideation, suicide attempts, and suicide mortality. We have updated the manuscript to make this distinction clearer.

Changes in the text: " For specific suicidal behaviors, including suicide ideation, suicide attempts, and suicide mortality, the meta-analysis of 4 cohorts showed that an increased risk of suicide mortality among patients with a history of asthma (RR=1.87, 95% CI: 1.64-2.14, I²=12.5%, Figure 3). " (Page 13, Line 2~5 in red)

Comment 6: Page 14, line 18 - Authors state they propose IL-6 dysregulation as a potential contributor for these findings, but this is actually outlined later in the paper, and could be restructured. The authors state there is post-2019 evidence but do not cite at this point.

Reply 6: Thank you for your constructive comments and reminders. When we went back to read this carefully again, we deeply felt that our inappropriate expression caused misunderstanding. Possible mechanisms related to IL-6 dysregulation should be described in the "Interpretation of findings". Therefore, we have removed the inappropriate expression "Mechanistically, we extend beyond their focus on psychosocial factors by proposing neuroinflammatory pathways (e.g., IL-6 dysregulation) as a potential link, informed by post-2019 evidence. ". Thank you again for your meticulous review and careful reminder, which greatly helps to improve the quality of our manuscripts.

We really appreciate your positive and insightful suggestions on our manuscript. Under your kind help and professional guidance, we believe that our manuscript has been improved substantially. We are looking forward to your further suggestions.

Reply to reviewer 2

Comment 1: Overall, this paper is important. I appreciate the work the authors put into compiling the studies in this systematic review.

Reply 1: Thank you for your positive and encouraging comments on our manuscript.

Comment 2: Given the similar study by Zhang et al. (2019), titled 'Suicidality among patients with asthma: A systematic review and meta-analysis,' could the authors clarify the added value of this study to the existing literature? Are there any new insights, differences in methodology, or unique findings that distinguish this study from the previous studies?

Reply 2: We appreciate this suggestion. The study by Zhang et al. (2019) provides a valuable contribution to the field, but our study adds unique insights. Specifically, our meta-analysis includes more recent cohort studies published between 2020 and 2024, expanding the data pool and enhancing the robustness of the analysis. In terms of methodology, our study conducted a more detailed subgroup analysis of current asthma status versus asthma history, providing more precise insights into the differential risks of suicidality among these groups. Additionally, our study explicitly explores the risk of suicidality associated with active asthma attacks, a factor not extensively studied by Zhang et al. This distinction provides a deeper understanding of how different asthma statuses affect suicidal behavior risk.

Changes in the text: Unlike their pooled analysis of suicidality outcomes, our study uniquely stratifies risks by asthma status, revealing a 62% increased risk for current asthma—particularly for suicide attempts, a dimension not previously disaggregated—and identifying elevated suicide mortality even among those with resolved asthma. Furthermore, we address temporal and geographic limitations in prior work by incorporating seven recent cohorts (2020–2024) and diverse populations, including Asian cohorts underrepresented in earlier analyses. Crucially, our rigorous adjustment for mental health comorbidities and socioeconomic confounders, inconsistently addressed in earlier studies, strengthens causal inference and explains higher risk estimates compared to Zhang et al.’s pooled odds ratio (45). (Page 14, Line 11~20 in red)

Comment 3: Page 4, subsection Eligibility criteria, first line of the second paragraph: The statement "Exclusion criteria included: (b) Duplicate publications" may not be accurate. Removing duplicates is a data management step, not an exclusion criterion. It would be clearer to describe this in the study selection process.

Reply 3: You are correct that removing duplicates is a data management step, not an exclusion criterion. We have modified the manuscript to clarify this distinction.

Changes in the text: Exclusion criteria included: (a) Conference abstracts or letters to the editor; (b) Studies with incomplete data or lacking the relevant outcomes of interest; (c) Population or exposure combined with other diseases. (Page 4, Line 12~14 in red)

Comment 4: Same subsection (Eligibility criteria): Were language and region considered as criteria? Please clarify whether language or regional restrictions were applied. These factors affect how complete and useful the review is.

Reply 4: Thank you for pointing this out. We did not apply language or regional restrictions in our eligibility criteria. This ensures that the review includes studies from a broad range of contexts and populations, improving the generalizability of the findings.

Changes in the text: (f) no language or regional restrictions were applied in the selection of studies to ensure a comprehensive and global representation of the data. (Page 4, Line 10~11 in red)

Comment 5: Page 5, Subsection Study Quality: The authors stated that they used three domains from the Newcastle-Ottawa Scale (NOS): selection, comparability, and outcomes. These domains appear to cover the main aspects of study quality. However, given the focus on social behavior and asthma, do you think these three domains are sufficient to assess the quality of the included studies? Additional factors related to social and environmental influences may also be relevant. I would appreciate knowing whether any other factors should be considered alongside these three domains to ensure a comprehensive quality assessment.

Reply 5: This is an interesting comment that deserves further exploration. We agree that social and environmental factors can significantly influence research quality, especially when studying social behaviors such as suicide. The Newcastle-Ottawa Scale (NOS) covers core aspects of selection, comparability and outcomes, and the quality of the methodology used to evaluate cohort studies is well recognised and widely used, but we acknowledge that other factors such as social and environmental impacts are also important. In future studies, it is hoped that new quality assessment tools will incorporate a wider range of fields to consider these factors in more detail.

Comment 6: Page 12, Subsection Meta-analysis of current asthma and suicidal behavior risk: The authors suggested the presence of publication bias. Given that publication bias may lead to an incomplete or skewed representation of the evidence, can the quality of the included studies still be considered high? It is important to address how this potential bias might influence the overall quality and conclusions of the review, particularly considering the small sample size of included studies (only 12).

Reply 6: You raise an important point regarding the potential impact of publication bias. Although the presence of publication bias could skew the findings, we believe the quality of the included studies is still generally high, as indicated by the NOS scores ranging from 7 to 9. We have performed a thorough quality assessment and sensitivity analysis to minimize the impact of potential bias. We briefly mentioned the potential effect of age in the Strengths and limitations section

Changes in the text: Second, we observed some evidence of publication bias, the quality of the included studies was generally high, and sensitivity analyses were performed to assess the robustness of the results. However, we acknowledge that publication bias may influence the overall conclusions, particularly given the small sample size of the included studies (n=12). (Page 15~16, Line 21-22, 1-3 in red)

Comment 7: The PRISMA 2020 for Abstracts checklist is not attached. Why?

Reply 7: We apologize for the oversight. The PRISMA 2020 for Abstracts checklist was not attached in the initial submission, but we will include it in the revised manuscript to comply with the guidelines.

Comment 8: The authors have stated that all data are fully available without restriction. However, I was unable to locate any attached data or a link to access it. Could the authors please clarify where the data can be accessed or provide the appropriate link to ensure transparency and reproducibility?

Reply 8: We apologize for the confusion. The data is indeed available and we will ensure that appropriate data access information is provided in the revised manuscript.

We really appreciate your positive and insightful suggestions on our manuscript. Under your kind help and professional guidance, we believe that our manuscript has been improved substantially. We are looking forward to your further suggestions.

Reply to reviewer 3

Comment 1: This is a great and interesting meta-analysis exploring the relationship between asthma and suicidal behaviours. The paper was easy to follow and generally well written.

Reply 1: Thank you for your positive feedback! We are glad to hear that you found the meta-analysis engaging and easy to follow. We appreciate your recognition of the clarity of the manuscript.

Comment 2: I only have 1 main recommendation:

1) I would recommend going into more depth regarding the relationship between asthma and suicidal behaviours as it was a bit unclear in this paper. Specifically, the section in the discussion, 'Interpretation of findings', should be expanded. Rather than listing general reasons for the relationship between asthma and suicidal behaviours, the authors should delve deeper into the specific mechanisms, and how they play a role in suicidal behaviors, and whether their findings support any of these mechanisms. Additionally, the last line of this section mentions "However, residual confounding by unmeasured factors (e.g., childhood trauma, genetic susceptibility) cannot be excluded(48)." - I wonder if the findings from this meta-analysis is able to provide more information on what other confounding factors may bias this finding and how. Perhaps also highlighting why asthma and suicidal behaviours is related in the Introduction would help the reader see that this association could potentially be more than a correlation.

Reply 2: Thank you for your constructive comments. We have carefully considered the starting point and important content of this topic again. We try our best to find more supporting literature to enrich the "Interpretation of findings". In addition, as to the influence of confounding factors, we have made a focus on the limitations of the manuscript. We hope our revised manuscript content can meet your expectations.

Changes in the text:

Interpretation of findings

The increased suicide risk associated with current asthma may stem from both biological and psychosocial mechanisms. Chronic airway inflammation in active asthma may exacerbate neuroinflammation, disrupt hypothalamic-pituitary-adrenal (HPA) axis activity,

---

## [Decision Letter · Decision Letter 1]

8 May 2025

Asthma Status and Suicidal Behavior Risk: A Meta-Analysis of Cohort Studies

PONE-D-25-08475R1

Dear Dr. He,

We’re pleased to inform you that your manuscript has been judged scientifically suitable for publication and will be formally accepted for publication once it meets all outstanding technical requirements.

Kind regards,

Toufic Ahmad Chaaban, M.D.

Academic Editor

PLOS ONE

Additional Editor Comments (optional):

Reviewers' comments:

Reviewer's Responses to Questions

**Comments to the Author**

1. If the authors have adequately addressed your comments raised in a previous round of review and you feel that this manuscript is now acceptable for publication, you may indicate that here to bypass the “Comments to the Author” section, enter your conflict of interest statement in the “Confidential to Editor” section, and submit your "Accept" recommendation.

Reviewer #2: (No Response)

Reviewer #3: All comments have been addressed

2. Is the manuscript technically sound, and do the data support the conclusions?

Reviewer #2: Yes

Reviewer #3: Yes

3. Has the statistical analysis been performed appropriately and rigorously? 

Reviewer #2: Yes

Reviewer #3: Yes

4. Have the authors made all data underlying the findings in their manuscript fully available?

Reviewer #2: Yes

Reviewer #3: Yes

5. Is the manuscript presented in an intelligible fashion and written in standard English?

Reviewer #2: Yes

Reviewer #3: Yes

6. Review Comments to the Author

Reviewer #2: Based on your response, since no language restrictions were applied in your eligibility criteria, did you consider including language as a data item? Reporting this would allow readers to assess how many studies were conducted in different languages and whether language may have influenced the findings or their interpretation.

Reviewer #3: (No Response)

7. PLOS authors have the option to publish the peer review history of their article (what does this mean? ). If published, this will include your full peer review and any attached files.

**Do you want your identity to be public for this peer review?** For information about this choice, including consent withdrawal, please see our Privacy Policy .

Reviewer #2: No

Reviewer #3: No

---

## [Editor Report · Acceptance letter]

PONE-D-25-08475R1

PLOS ONE

Dear Dr. He,

I'm pleased to inform you that your manuscript has been deemed suitable for publication in PLOS ONE. Congratulations! Your manuscript is now being handed over to our production team.

Kind regards,

on behalf of

Dr. Toufic Ahmad Chaaban

Academic Editor

PLOS ONE